# Total infectome characterization of respiratory infections in pre-COVID-19 Wuhan, China

**Mang Shi**[1,2☯¤], **Su Zhao**[3☯], **Bin Yu**[4☯], **Wei-Chen Wu**[1☯¤], **Yi Hu**[3☯], **Jun-Hua Tian**[4], **Wen Yin**[3], **Fang Ni**[3], **Hong-Ling Hu**[3], **Shuang Geng**[3], **Li Tan**[3], **Ying Peng**[4], **Zhi-Gang Song**[1], **Wen Wang**[1,5], **Yan-Mei Chen**[1], **Edward C. Holmes** ID[1,2], **Yong-Zhen Zhang** ID[1] *

**1** Zhongshan Hospital, State Key Laboratory of Genetic Engineering, School of Life Sciences, Human Phenome Institute and Shanghai Public Health Clinical Center, Fudan University, Shanghai, China, **2** Sydney Institute for Infectious Diseases, School of Life and Environmental Sciences and School of Medical Sciences, The University of Sydney, Sydney, Australia, **3** Department of Pulmonary and Critical Care Medicine, The Central Hospital of Wuhan, Tongji Medical College, Huazhong University of Science and Technology, Wuhan, China, **4** Wuhan Center for Disease Control and Prevention, Wuhan, China, **5** Department of Zoonosis, National Institute for Communicable Disease Control and Prevention, China Center for Disease Control and Prevention, Beijing, China

☯ These authors contributed equally to this work.
¤ Current address: School of Medicine, Sun Yat-sen University, Guangzhou, China
* zhangyongzhen@fudan.edu.cn

**Data Availability Statement:** All non-human reads have been deposited in the SRA databases under the project accession PRJNA699976. Relevant consensus virus genome sequences have been

## Abstract

At the end of 2019 Wuhan witnessed an outbreak of "atypical pneumonia" that later developed into a global pandemic. Metagenomic sequencing rapidly revealed the causative agent of this outbreak to be a novel coronavirus denoted SARS-CoV-2. To provide a snapshot of the pathogens in pneumonia-associated respiratory samples from Wuhan prior to the emergence of SARS-CoV-2, we collected bronchoalveolar lavage fluid samples from 408 patients presenting with pneumonia and acute respiratory infections at the Central Hospital of Wuhan between 2016 and 2017. Unbiased total RNA sequencing was performed to reveal their "total infectome", including viruses, bacteria and fungi. We identified 35 pathogen species, comprising 13 RNA viruses, 3 DNA viruses, 16 bacteria and 3 fungi, often at high abundance and including multiple co-infections (13.5%). SARS-CoV-2 was not present. These data depict a stable core infectome comprising common respiratory pathogens such as rhinoviruses and influenza viruses, an atypical respiratory virus (EV-D68), and a single case of a sporadic zoonotic pathogen–*Chlamydia psittaci*. Samples from patients experiencing respiratory disease on average had higher pathogen abundance than healthy controls. Phylogenetic analyses of individual pathogens revealed multiple origins and global transmission histories, highlighting the connectedness of the Wuhan population. This study provides a comprehensive overview of the pathogens associated with acute respiratory infections and pneumonia, which were more diverse and complex than obtained using targeted PCR or qPCR approaches. These data also suggest that SARS-CoV-2 or closely related viruses were absent from Wuhan in 2016–2017.

deposited in NCBI/GenBank under the accessions MW567157- MW567162, MW570805-MW570808, MW571087- MW571107, MW587035- MW587095.

**Funding:** This study was supported by the National Natural Science Foundation of China 32041004 (YZZ), 31930001 (YZZ), 32130002 (YZZ), 81861138003 (YZZ) and 81672057 (YZZ). E.C.H. is supported by an ARC Australian Laureate Fellowship (FL170100022). The funders had no role in study design, data collection and analysis, decision to publish, or preparation of the manuscript.

**Competing interests:** The authors have declared that no competing interests exist.

## Author summary

Acute respiratory infections and pneumonia are a significant public health concern on a global scale. This was brought sharply into focus by the emergence of COVID-19 at the end of 2019 in Wuhan, China. Far less is known, however, about the total "infectomes" associated with respiratory infection and pneumonia: that is, all the viruses, bacteria and fungi that might be associated with these important human diseases. Herein, we describe total infectomes in bronchoalveolar lavage fluid samples taken from 408 patients presenting with pneumonia and acute respiratory infection in a single hospital in Wuhan between 2016 and 2017. From these samples we identified 35 pathogen species, comprising 13 RNA viruses, 3 DNA viruses, 16 bacteria and 3 fungi, often at high abundance and including multiple co-infections. We found no evidence for SARS-CoV-2. These data reveal a stable core infectome comprising common respiratory pathogens as well as their history of global transmission.

## Introduction

The emergence of COVID-19 at the end of 2019 has had a profound impact on the world. The causative agent, a betacoronavirus (*Coronaviridae*) termed SARS-CoV-2, has high transmissibility and causes mild to severe respiratory symptoms in humans [1–3]. The first documented case of SARS-CoV-2 was reported in Wuhan, Hubei province, China[3]. Despite intensive research into SARS-CoV-2 and COVID-19, important questions remain regarding the emergence of this virus, including for when and where it appeared and how long it was circulating in human populations prior to its initial detection in December 2019.

Meta-transcriptomics has several advantages over traditional diagnostic approaches based on serology or PCR [4]: it targets all types of micro-organisms simultaneously, identifies potential pathogens without *a priori* knowledge of what micro-organisms might be present, reveals the information (RNA) expressed by the pathogen during infection that is central to agent identification and studies of disease association. This method has been proven highly successful in revealing the entire virome and microbiome in a diverse range of species [5–8], including the initial identification of SARS-CoV-2 from patients with severe pneumonia [1].

Acute respiratory infections and pneumonia are a significant public health concern on a global scale. However, far less is known about the total infectomes associated with respiratory infection and pneumonia. The recent application of mNGS approaches have revealed a diverse range of pathogens and outlined the infectomes potentially associated with severe pneumonia [9], lower respiratory tract infection [10], and other respiratory diseases [8,11]. Herein, we report the total infectome surveillance of 408 patients presenting with pneumonia and acute respiratory infections at Wuhan Central Hospital in 2016 and 2017 and so prior to the emergence of SARS-CoV-2. The purpose of this study was to use an un-biased meta-transcriptomics tool to characterize the total infectome within these patients. In addition, since the sampling took place before the outbreak of COVID-19, this study represents an opportunity to characterize the entire range of pathogens within a cohort and determine the microbial composition of the human population in which SARS-CoV-2 was initially reported.

## Results

### Patient context

We considered 408 patients clinically diagnosed with pneumonia or acute respiratory infection at the Central hospital of Wuhan in Wuhan, China. The sampling period lasted for 20 months and covered the period between May 2016 to December 2017, two years before the onset of the COVID-19 pandemic (Fig 1A). The male-to-female ratio among the patients sampled was 1.4 (Fig 1B), with age ranging from 16 to 90 years (medium, 62). Pre-existing medical conditions present in these patients included hypertension (n = 108), diabetes (n = 46), bronchiectasia (n = 31), chronic obstructive pulmonary disease (COPD, n = 23), cancer (n = 12), and heart disease (n = 10). Based on evaluations made by clinicians at the hospital, 27 patients were described as severely ill, with 381 presenting with non-severe syndromes (Fig 1E). The mortality for the entire cohort was 0.74% (n = 3) and the average duration of hospitalization was 8 days (range 2–322, medium 9). The average time between hospitalization and sample collection was 3.36 days (range 0–33, medium 3). No correlation was found between age, gender, severity, and sampling time. One exception was age and gender, with the male patients (range 16–90, medium 63) were slightly older than female patients (range 16–85, medium 59) (S1 Table).

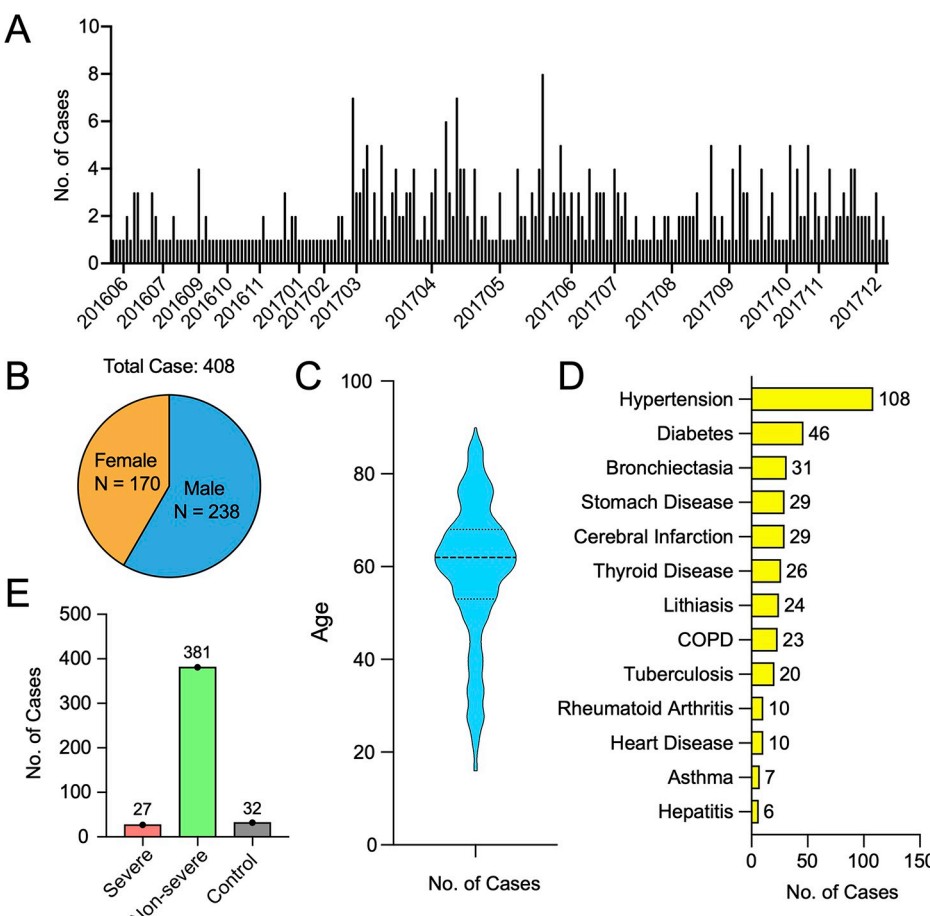

**Fig 1. Patient recruitment.** (A) Sampling frequency and intensity during the study period. (B) Male-to-female ratio of the recruited patients. (C) Age structure of all patients involved in this study depicted using violin plots. (D) Type and frequencies of pre-existing conditions. (E) The number of severe, non-severe and control cases.

## Total infectome

Meta-transcriptomic analysis of the BALF samples identified a wide range of RNA viruses, DNA viruses, bacteria and fungi. For the purposes of this study, we only characterized those likely associated with human disease (i.e., pathogens). This included: (i) existing species that are known to be associated with human disease, and (ii) potentially novel pathogens that have not been previously characterized. For the latter, we only considered DNA and RNA viruses that are related to a virus genus or family that have previously been shown to infect mammals. The abundance threshold for pathogen positives was set at 1 sequence read per million (or RPM). In addition, likely commensal bacteria were not considered here.

Based on these criteria we did not identify any potential novel viral pathogens. All the microbes identified belonged to those previously characterized as human pathogens, comprising 13 RNA viruses, 3 DNA viruses, 16 bacteria and 3 fungal pathogens (Fig 2). The case positive rate for all pathogens was 49.5% (n = 202, Fig 2A), many of which were only associated with RNA viruses (32.1%, n = 131) or bacteria (25.2%, n = 103). Co-infection with two different pathogen species was also commonplace, comprising a total of 55 (13.5%) cases (Fig 2A). Among the pathogens identified, most were common respiratory pathogens such as influenza viruses, rhinoviruses, *Pseudomonas aeruginosa* and *Haemophilus influenzae* (Fig 2B). In addition, we identified a number of unconventional respiratory pathogens that are often not included in respiratory pathogen screening panels but known to cause severe infections in the respiratory tract or lungs, including enterovirus D68 and *Chlamydia psittaci* (see below).

Finally, none of the pathogens described here appeared in the blank sequencing controls. An exception was *Escherichia coli*, which appeared in the experimental, healthy control and blank control groups (1296–2173 RPM), but was regarded as likely contamination and so not considered further. Since the blank control samples were generated using the same procedures for RNA extraction, library preparation and sequencing as the experimental groups, these results effectively exclude the possibility that the pathogens described above were of contaminant origin. Notably, none of age, gender, sampling time and disease severity of patients showed significant effect on the abundance of all identified pathogens (S1 Table).

## Viruses

RNA viral pathogens exhibited both great diversity (13 species) and abundance (up to 52% of total RNA) in the BALF samples examined here. The most frequently detected RNA viruses were human rhinoviruses A-C (HRV, n = 50), followed by influenza A virus (IAV, n = 29), human parainfluenza virus type 3 (HPIV3, n = 20), influenza B virus (IBV, n = 8), and human metapneumovirus (HPMV, n = 7) (Fig 2B). While the majority were found throughout the study period, some had more specific timescales (Fig 2C). For example, influenza B viruses were mostly identified in 2017, whereas enterovirus D (ENV-D, n = 6) was only detected in the summer of 2016. In addition, we identified all four types of common cold associated coronaviruses–OC43 (n = 4), HKU1 (n = 4), 229E (n = 6) and NL63 (n = 1)–all of which had a relatively low prevalence in our cohort. Importantly, none of the libraries contained any hit to SARS-CoV or SARS-CoV-2, a result confirmed by both read mapping and a blast analysis against the corresponding viral genomes.

In comparison to RNA viruses, the DNA viruses identified were limited in diversity and abundance. All three major types of human herpesviruses were identified–HSV1 (n = 3), CMV (n = 3), and EBV (n = 2)–although always at low abundance (up to 30 RPM or 0.003%) (Fig 2C). Another common DNA virus that causes respiratory disease—adenovirus—was also identified in several cases, although at abundance levels lower than the 1 RPM threshold such that it was considered a 'negative' result in this context.

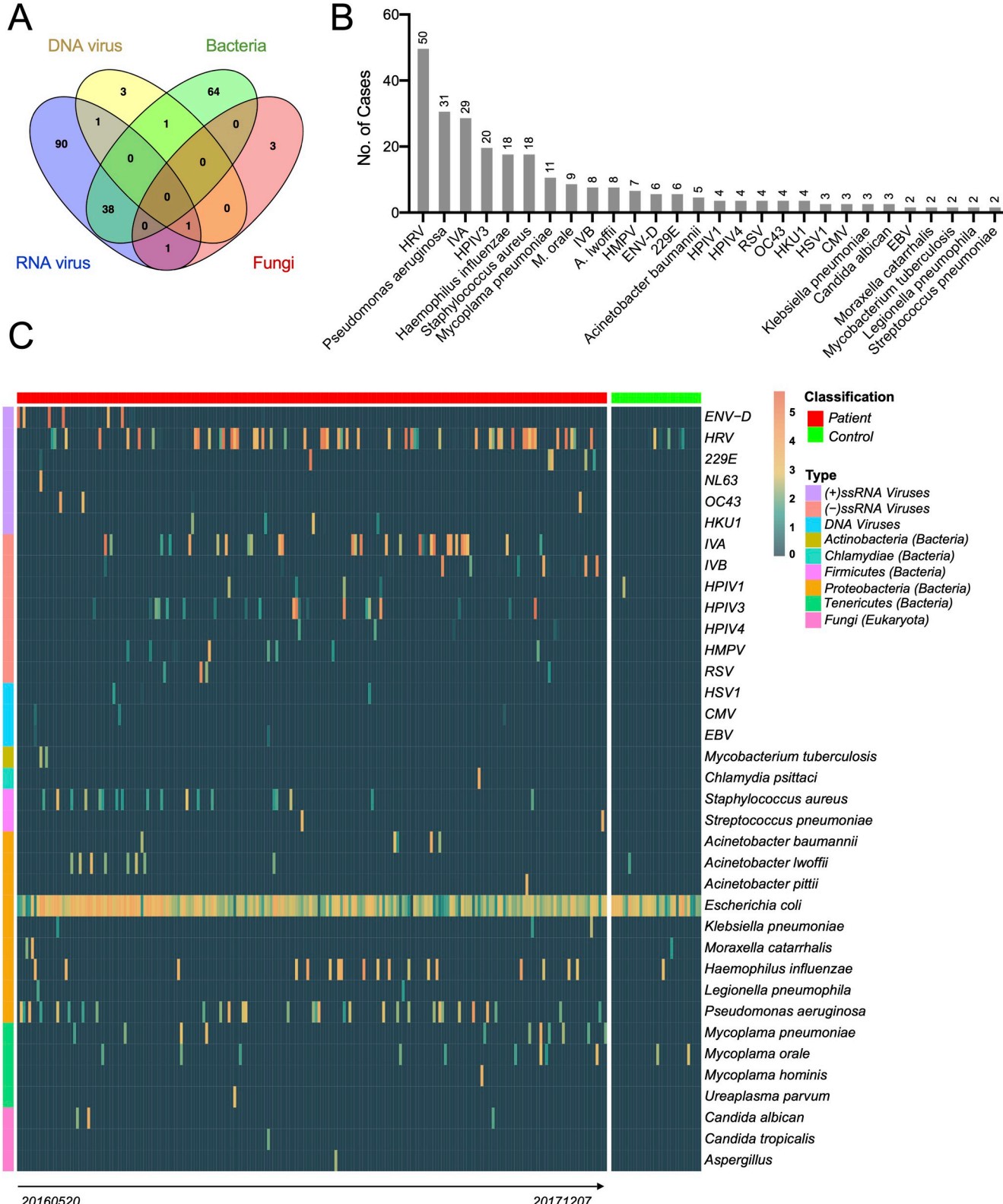

**Fig 2. Prevalence and abundance of viral, bacterial and fungal pathogens in the 408 cases examined in this study.** (A) Proportion of cases infected with RNA virus, DNA virus, bacteria, fungus and with mixed infections. (B) Prevalence of each pathogen, ordered by the number of cases. (C) Heat map showing the prevalence and abundance of pathogens in diseased and control samples. *Escherichia coli* is not regarded as pathogen but is shown here as an example of non-pathogen contamination. The samples (x-axis) are divided into "Patient" and "Control" groups, each ordered chronically. The pathogens (y-axis) are divided into four categories: RNA viruses, DNA viruses, bacteria, and fungi.

## Bacteria and fungi

The most common bacterial pathogens identified included *Pseudomonas aeruginosa* (n = 31), *Haemophilus influenzae* (n = 18), *Staphylococcus aureus* (n = 18), and *Mycoplasma pneumoniae* (n = 11), all of which are common respiratory pathogens. *Acinetobacter* bacteria were also prevalent in our cohort, including *Acinetobacter baumannii* (n = 5) and *A. pittii* (n = 1), both of which are commonly associated with hospital-acquired infections. Other important respiratory pathogens, such as *Mycobacterium tuberculosis* (n = 2), *Legionella peumophila* (n = 2), *Streptococcus pneumoniae* (n = 2), *Klebsiella pneumoniae* (n = 3) and *Moraxella catarrhalis* (n = 2), were also detected, although at a relatively low prevalence. Of particular interest was the identification of a single case of *Chlamydia psittaci*–a potentially bird-associated zoonotic pathogen–present in the BALF at relatively high abundance (6396 RPM). Conversely, all the fungal pathogens identified here–*Candida albican* (n = 3), *C. tropicalis* (n = 1), and *Aspergillus spp.* (n = 1)–were common pathogens known to cause respiratory infections.

## qPCR confirmation of pathogen presence and abundance

For each of the RNA viral pathogens identified here, we performed a qRT-PCR assay on positive samples to confirm their presence and validate their abundance level as measured using our meta-transcriptomic approach. Strong correlations were observed between the abundance measured by qPCR (i.e., CT value) and those estimated by read count after a log 2 conversion ($-0.8 <$ Pearson's $R < -1$, Fig 3). Hence, the quantification by the two methods is strongly comparable. Finally, the qPCR assays for SARS-CoV-2 provided negative results for all the samples examined here.

## In-depth phylogenetic characterization of pathogens

Although no novel viral pathogens were identified in this study, those viruses detected were characterized by substantial phylogenetic diversity, reflected in the presence of multiple viral lineages that highlight their complex epidemiological history in Wuhan (Fig 4). We identified more than 14 genomic types of rhinovirus A, 7 of rhinovirus B, and 4 of rhinovirus C. A similar pattern of the co-circulation of multiple viral lineages was observed in other viruses. For example, the influenza A viruses identified in this study can be divided into the H1N1 and H3N2 subtypes, each containing multiple lineages that clustered with viruses sampled globally and reflecting the highly connected nature of Wuhan (Fig 4).

## Pathogen presence and abundance in diseased and healthy individuals

Our meta-transcriptomic analysis revealed that many RNA viruses and bacteria detected were present at extremely high abundance levels ($>1\%$, and up to 52% of total RNA) and hence likely indicative of acute disease. This was particularly true of six species of RNA viruses–EV-D68, the influenza viruses, HRV, HPIV3, 229E –as well as two species of bacteria (*Haemophilus influenzae* and *Pseudomonas aeruginosa*) (Fig 5). Together, these comprise a total of 54 cases (13.2% of total diseases cases).

In marked comparison, high levels of abundance were never observed in the healthy control group (Fig 5). The highest abundance in this group was observed for *Haemophilus influenzae* at 1302 RPM (0.13% of total RNA). Indeed, for most pathogens, the abundance level in healthy group was either undetectable or well below that observed in the diseased group, with the exception of Human parainfluenza virus type 1 (HPIV) and *Mycoplasma orale* for which the abundance levels were higher in the control group, although the sample size for both pathogens was relatively small.

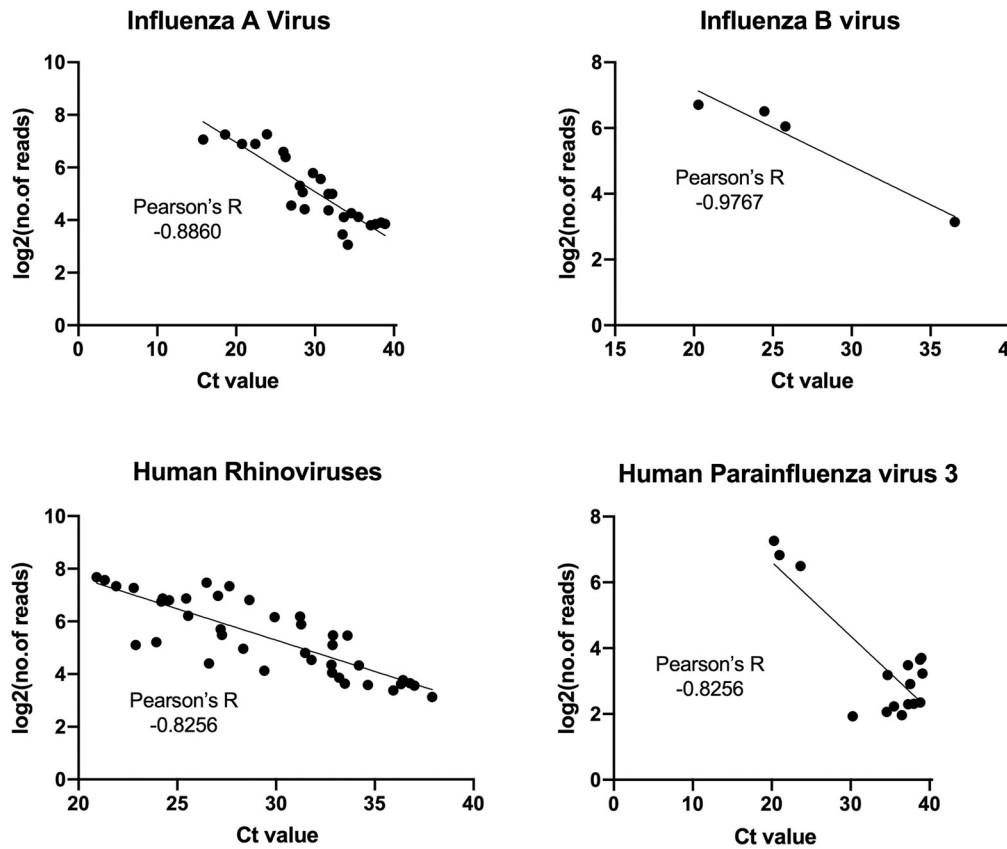

**Fig 3. Comparisons of pathogen abundance measured by qPCR and meta-transcriptomics approaches.** Abundance by qPCR methods is measured by cycle threshold, or CT value, while those obtained by meta-transcriptomics are measured by "log2(number of reads)". The comparisons are performed on four most abundant pathogens: influenza A virus, influenza B virus, human rhinoviruses, and human parainfluenza virus 3. Pearson's correlation coefficient between CT values and log2(number of reads) is estimated for each pathogen.

## Discussion

Our study provides a critical snapshot of the respiratory pathogens present in Wuhan prior to the emergence of SARS-CoV-2. Our unbiased metagenomic survey in patients presenting with pneumonia or acute respiratory infection provides strong evidence that SARS-CoV-2 or any related SARS-like viruses were absent in Wuhan approximately two years prior to the onset of pandemic, although a variety of common cold coronaviruses (HKU1, OC43, 229E, and NL63) were commonly detected in our cohort. Indeed, the earliest COVID-19 case, identified by qRT-PCR or next-generation sequencing-based assays performed at designated authoritative laboratories, can currently only be traced back to December 2019 in Wuhan [12]. In addition, a retrospective survey of 640 throat swabs from patients with influenza-like illness in Wuhan from the period between 6 October 2019 and 21 January 2020 did not find any evidence of SARS-CoV-2 infection prior to January 2020 [13], such that the ultimate origin of SARS-CoV-2 remains elusive [14].

The data presented provide a comprehensive overview of the infectome associated with pneumonia or acute respiratory infections in Wuhan, which is clearly more diverse and complex than described using previous surveys based on targeted PCR or qPCR approach alone [15]. In light of these observations, we can divide the respiratory infectome into three major groups based on epidemiological characteristics: (i) the "core" infectome that is commonly

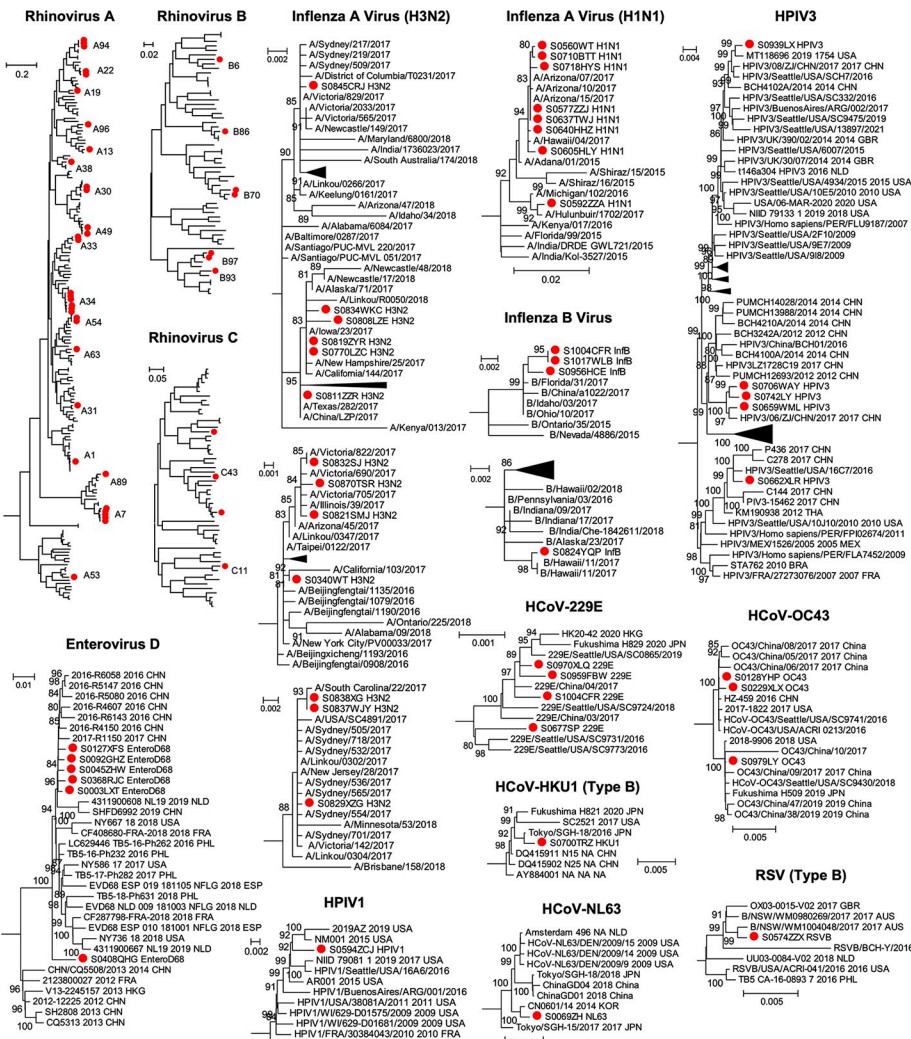

**Fig 4. Evolutionary relationships of RNA virus pathogens identified in this study.** Phylogenetic trees of each virus were estimated using the maximum likelihood method implemented in PhyML. Sequences identified from this study were marked with red solid circle. For larger trees, we only show the lineages or sub-lineages that contain sequences identified in this study.

found in patients with respiratory infection and is expected to occur globally each year, (ii) an "emerging" infectome present during outbreaks but which are not typically found in the geographic regions under investigation, and (iii) the sporadic presence of new or rare pathogens including those of zoonotic origin.

The core infectome comprised a wide range of common respiratory or systemic pathogens that are subject to frequent screening in hospitals. These include influenza viruses, HMPV, RSV, *Moraxella catarrhalis*, *Acinetobacter spp.*, *Klebsiella pneumoniae*, *Mycoplasma spp.*, *Haemophilus influenzae*, *Pseudomonas aeruginosa*, *Staphylococcus aureus* and *Streptococcus pneumoniae*. However, the remaining pathogens identified here, including rhinoviruses, parainfluenza viruses, coronaviruses, and herpesviruses, have often received far less attention from clinicians and are sometimes ignored entirely, most likely due to the lack of association with severe disease in adults [16–18]. Nevertheless, our results showed these "neglected" respiratory viruses had high diversity, abundance and prevalence in the cohort of pneumonia or

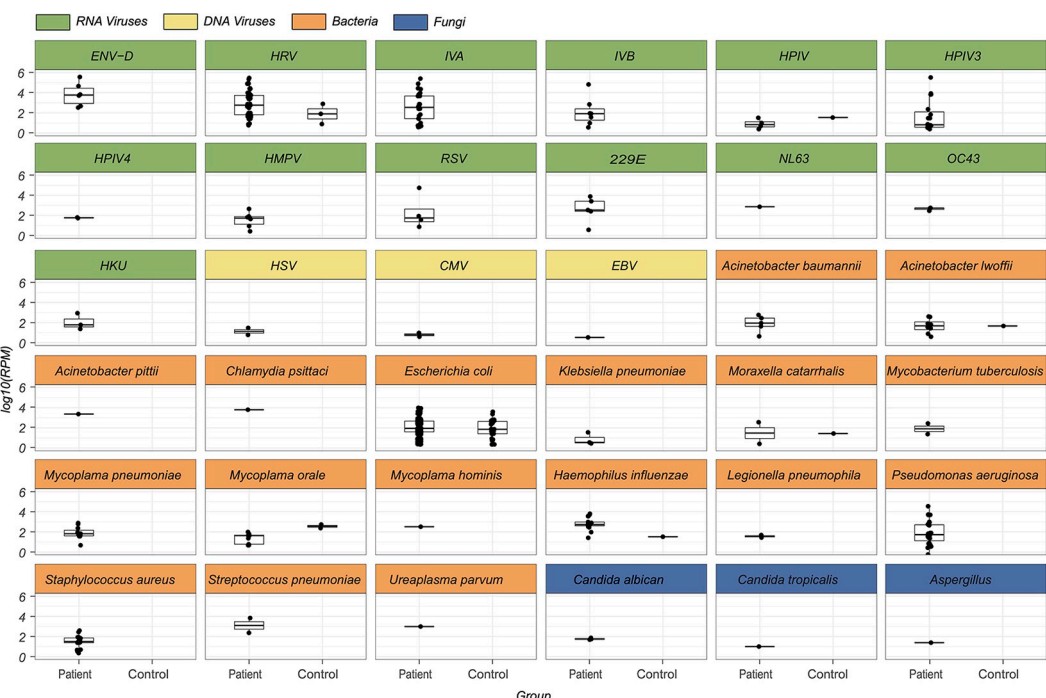

**Fig 5. Comparison of prevalence levels between the healthy and control groups.** Boxplots for patient and control groups for each pathogen identified in this study, including RNA virus (green), DNA virus (yellow), bacteria (orange) and fungus (blue). For clarity, only non-zero abundance levels are reported.

acute respiratory patients studied here in comparison to healthy controls, such that their role as agents of disease should not be underestimated. One scenario is that they represent opportunistic pathogens that take advantage of weakened immunity, such as herpesviruses associated acute respiratory distress (ARDS) [19]. It is also possible that their pathogenic effects have yet to be identified and may extend to disease manifestations beyond respiratory infections. For example, deep sequencing of a brain biopsy sample suggested that coronavirus OC43 may sometimes be associated with fatal encephalitis in humans [20].

We also identified a potential emerging infectome, in this case comprising a single virus– EV-D68 –that may represent a regional or national outbreak of an unconventional respiratory pathogen. The prevalence of EV-D68 remained low from its discovery in 1967 until 2014 [21], when a major outbreak started in the United States and spread to more than 20 countries [22], causing severe respiratory illness with potential neurological manifestations such as acute flaccid paralysis in children [23,24]. In China, EV-D68 was only sporadically reported [25,26], although serological surveys suggest a much wider prevalence for both children and adults since 2009 [27]. We identified six EV-D68 cases, all adults that presented within a relatively narrow time window between June and December 2016. Phylogenetic analysis revealed that the sequences of these viruses were closely related to each other and to other Chinese strains from the same period (Fig 4), suggesting that it may be a part of a larger outbreak in China. The EV-D68 cases identified here showed moderate to severe respiratory symptoms, although viral abundance was generally very high, with four of six cases showing >106 RPM (i.e., >0.01% of total RNA) in the BALF sample. This highlights the active replication and massive proliferation of viruses within the respiratory system of these patients.

Finally, our zoonotic infectome also comprised a single pathogen, *Chlamydia psittaci*, that is associated with avian species but which causes occasional outbreaks in domestic animals

(i.e., pigs, cattle, and sheep) and humans [28]. In humans, *C. psittaci* infections often starts with influenza-like symptoms but can develop into serious lung infections and even death [29]. The single case of *C. psittaci* identified here was at relatively high abundance level (6396 RPM) and caused a relatively severe disease, with the patient experiencing expiratory dyspnea, severe pneumonia and pleural effusion, and was subsequently transferred to an intensive care unit (ICU) for further treatment. Since the patient had no travel history for a month prior to illness, this discovery underlines the risk of local exposure to this bacterium.

Our phylogenetic analysis of the metagenomic data generated revealed extensive intra-specific diversity in each virus species identified highlights the complex epidemiological history of these pathogens. Indeed, the influenza A viruses (H1N1 and H3N2), influenza B virus, HPIV3 and HCoV-OC43 discovered here all comprised multiple lineages (Fig 4), suggesting these viruses were introduced from diverse sources. Since some of the viruses were closely related to those circulating in other countries it is possible that they represent overseas importations: this is not surprising given that Wuhan is a major domestic travel hub and well linked internationally. Indeed, the rapid and widespread transmission of SARS-CoV-2 between Wuhan and other major cities globally was key to seeding the global pandemic of COVID-19, with early cases in a number of localities all linked to travel from Wuhan [29–32].

With the popularization of next-generation sequencing platform in major hospitals, the meta-transcriptomic approach outlined here can be easily integrated into diagnostic practice with much greater speed and significantly more information output than traditional technologies, providing a broad-scale understanding of infectious disease in general.

## Material and methods

### Ethics statement

The sampling and experimental procedures for this study were reviewed by the ethics committees of the Central Hospital of Wuhan and the National Institute for Communicable Disease Control and Prevention, Chinese Center for Disease Control and Prevention. Written informed consents were taken from all patients and volunteers recruited in this study. In addition, for child patients written consents were also obtained from their parents or guardians. Physicians were informed of results of the pathogen discovery exercise as soon as the meta-transcriptomic results were obtained.

### Sample collection from patients and controls

More than 1000 patients were recruited from the Central Hospital of Wuhan between 2016 and 2017. The target clinical conditions were community-acquired pneumonia and acute respiratory infection based on the initial diagnosis made by clinicians. All patients were hospitalized and subject to bronchoalveolar lavage fluid (BALF) collection required by the initial diagnosis for pneumonia or acute respiratory distress syndrome and independent of this study. The BALF sample was divided into two parts for the clinical laboratory test and this study, respectively. Of the BALF samples collected, 408 were subjected to meta-transcriptomic analysis based on their condition and the time between hospitalization and sample collection. No diagnostic information was provided prior to sample selection. To establish a healthy control group, 5ml-10ml of BALF samples were also taken from 32 volunteers without respiratory symptoms between March 27th to June 15th, 2017. We also included 10 blank controls where only RNase free water was used for nucleic acid extraction and library construction, although only four of these produced viable RNA sequencing results.

## Meta-transcriptomic pathogen discovery pipeline

We followed a standard protocol for meta-transcriptomics analysis for each BALF sample. Total RNA was first extracted from 200–300ul of each sample using the RNeasy Plus Universal Kit (Qiagen, USA) according to the manufacturer's instructions. From the extracted RNA, we performed human rRNA removal and low concentration library construction procedures with the Trio RNA-Seq kit (NuGEN Technologies, USA). The libraries were then subjected to 150bp pair-end sequencing on an Illumina HiSeq 4000 platform at Novagene (Beijing), with target output of 10G base pairs per library. For each of the sequencing results generated, we removed adaptor sequences, non-complex reads, as well as duplicated reads using the BBmap software package. Human and ribosomal RNA (rRNA) reads were subsequently removed by mapping the de-duplicated reads against the human reference genome (GRCh38/hg38) and the comprehensive rRNA sequence collection downloaded from the SILVA database [33].

The remaining sequencing reads were subject to a pathogen discovery pipeline. For virus identification, sequence reads were directly compared against reference virus databases using the blastn program (version 2.9.0) and against the non-redundant protein (nr) database using Diamond blastx (version 0.9.25) [34], with an e-value threshold set at 1E-10 and 1E-5 for blastn and Diamond blastx analyses, respectively. Viral abundance was summarized from both analyses, calculated using the relation: total viral reads/total non-redundant reads*1 million (i.e., reads per million of total non-redundant reads, RPM). To identify highly divergent virus genomes, reads were assembled using Megahit (version 1.1.3) [35] into contigs before comparison against the non-redundant nucleotide (nt) and nr databases. Those reads with <90% amino acid similarity to known viruses were treated as potential novel virus species. For bacterial and fungi identification, we first used MetaPhlAn (version 2) [36] to identify potential species in both groups. Relevant background bacterial and fungal genomes were subsequently downloaded from NCBI/GenBank and used as a template for read mapping using Bowtie2 (version 2.3.5.1) [37]. Based on the mapping results for each case, we generated relevant contigs for blastn analyses against the nt database to determine taxonomy to the species level. The abundance level of bacterial and fungal pathogens was also calculated in the form of RPM based on genome and mitochondrial genome read counts, respectively.

A microbe was considered as "positive" within a specific sample if its abundance was greater than 1 RPM. To prevent false positives resulting from index hopping, we used a threshold of 0.1% for viruses present in the same sequencing lane: that is, if the libraries contain less than 0.1% of the most abundant library it is treated as "negative".

## Confirmatory testing by conventional methods

For RNA viral pathogen positive samples, the same sample that was positive for a specific pathogen was also subjected to a qRT-PCR assay with primers sets designed for this or related group of pathogens (S2 Table). In addition, the qRT-PCR assays using primers targeting the ORF1ab and N genes of SARS-CoV-2 were performed on these samples using a commercial COVID-19 nucleic acid detection kit (DaAn Gene, China) (Cat. DA0932). RNA was first reverse transcribed by SuperScript III First-Strand Synthesis SuperMix for qRT-PCR (Invitrogen, California), and then amplified by TaqPath ProAmp Master Mix (Applied Biosystems, California). A cycle threshold (CT) value of 38 and above was treated as negative.

## Pathogen genomic analyses

For viruses at high abundance levels (i.e., > 1000 RPM), complete genomes were assembled using Megahit and confirmed by mapping reads against the assembled contigs. To choose corresponding reference sequences for phylogenetic analyses, all viral genomic data were first

downloaded from NCBI/GenBank under a specific taxonomic group (species or family) and clustered based on genetic similarities (99% ~ 99.9% based on virus species). We then selected those sequences that were (i) representative of background genetic diversity from each of the clusters and (ii) the closest hits in the blast analyses to establish a reference data set. These genomes were then aligned with related reference virus sequences downloaded from NCBI/GenBank using MAFFT(version 7.490) [38]. Ambiguously aligned regions were removed using Trimal (version 1.2) [39]. Phylogenetic trees of each data set were estimated using the maximum likelihood approach implemented in PhyML (version 3.0) [40], employing the GTR model of nucleotide substitution and SPR branch swapping. The support for each node in the tree was estimated using an approximate likelihood ratio test (aLRT) employing the Shimodaira-Hasegawa-like procedure.

## Supporting information

**S1 Table. Correlation among unordered categorical variables and pathogen abundance.** (XLSX)

**S2 Table. Primers used for RNA virus confirmation in qRT-PCR assays.** (XLSX)

## Acknowledgments

We sincerely acknowledge Ms. Yuan-Yuan Pei and Mr. Yu-Yi Zhang for performing qPCR assays to screen SARS-CoV-2, and Mr. Shu-Jian Hu for his contribution to draw the Striking Image.

## Author Contributions

**Conceptualization:** Yong-Zhen Zhang.

**Data curation:** Mang Shi, Wei-Chen Wu, Yan-Mei Chen, Yong-Zhen Zhang.

**Formal analysis:** Mang Shi, Wei-Chen Wu, Yan-Mei Chen.

**Funding acquisition:** Yong-Zhen Zhang.

**Investigation:** Su Zhao, Bin Yu, Yi Hu, Jun-Hua Tian, Wen Yin, Fang Ni, Hong-Ling Hu, Shuang Geng, Li Tan, Ying Peng, Zhi-Gang Song, Wen Wang.

**Methodology:** Mang Shi, Wei-Chen Wu, Wen Wang.

**Project administration:** Yong-Zhen Zhang.

**Resources:** Su Zhao, Bin Yu, Yi Hu, Jun-Hua Tian, Wen Yin, Fang Ni, Hong-Ling Hu, Shuang Geng, Li Tan, Ying Peng.

**Supervision:** Edward C. Holmes, Yong-Zhen Zhang.

**Validation:** Wei-Chen Wu, Zhi-Gang Song.

**Visualization:** Mang Shi, Yan-Mei Chen.

**Writing – original draft:** Mang Shi, Su Zhao, Bin Yu, Wei-Chen Wu, Yi Hu, Jun-Hua Tian, Wen Yin, Fang Ni, Hong-Ling Hu, Shuang Geng, Li Tan, Ying Peng, Zhi-Gang Song, Wen Wang, Yan-Mei Chen, Edward C. Holmes, Yong-Zhen Zhang.

**Writing – review & editing:** Mang Shi, Su Zhao, Bin Yu, Wei-Chen Wu, Yi Hu, Jun-Hua Tian, Wen Wang, Yan-Mei Chen, Edward C. Holmes, Yong-Zhen Zhang.

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
