## [Decision Letter · Decision Letter 0]

15 Oct 2021

Dear Dr. Zhang,

Thank you very much for submitting your manuscript "Total Infectomes Characterization of Respiratory Infections in pre-COVID-19 Wuhan, China" for consideration at PLOS Pathogens. As with all papers reviewed by the journal, your manuscript was reviewed by members of the editorial board and by several independent reviewers. In light of the reviews (below this email), we would like to invite the resubmission of a significantly-revised version that takes into account the reviewers' comments.

Your manuscript has been reviewed by four experts in the field. The Reviewers ##2-4 outlined major issues concerning the use of qPCR and NGS, as well as the selection of viruses for the phylogenetic analysis. The Reviewer #2 asked about the comparable analysis of the BALF samples collected after 2017 and before the onset of the COVID-19 pandemic, which belong to the manuscript scope and are critical to establish the baseline of respiratory infections in the Wuhan area. Also, the Reviewers ##1-2,4 raised other concerns and provided thoughtful suggestions which we believe will help you improving the manuscript and we urge you to follow carefully.

We cannot make any decision about publication until we have seen the revised manuscript and your response to the reviewers' comments. Your revised manuscript is also likely to be sent to reviewers for further evaluation.

Sincerely,

Alexander E. Gorbalenya, PhD, DSci

Associate Editor

PLOS Pathogens

Marco Vignuzzi

Section Editor

PLOS Pathogens

Kasturi Haldar

Editor-in-Chief

PLOS Pathogens

orcid.org/0000-0001-5065-158X

Michael Malim

Editor-in-Chief

PLOS Pathogens

orcid.org/0000-0002-7699-2064

Your manuscript has been reviewed by four experts in the field. The Reviewers ##2-4 outlined major issues concerning the use of qPCR and NGS, as well as the selection of viruses for the phylogenetic analysis. The Reviewer #2 asked about the comparable analysis of the BALF samples collected after 2017 and before the onset of the COVID-19 pandemic, which belong to the manuscript scope and are critical to establish the baseline of respiratory infections in the Wuhan area. Also, the Reviewers ##1-2,4 raised other concerns and provided thoughtful suggestions which we believe will help you improving the manuscript and we urge you to follow carefully.

Reviewer's Responses to Questions

**Part I - Summary**

Reviewer #1: The manuscript by Shi et al presents the viruses, bacteria, and fungi that can be found in acute respiratory infections / pneumonia patients living in Wuhan area in the years 2016 and 2017. The NGS data on the infecting pathogens is presented, the “infectome”. Several pathogens were found, yet no SARS-CoV-like viruses. The absence of SARS-CoVs confirms earlier reports that PCR-checked samples collected in 2019 in Wuhan (Kong et al . https://www.nature.com/articles/s41564-020-0713-1 ). The manuscript provides a clear presentation of the pathogens that did play a prominent or a less prominent role in the respiratory infections. The study is clear and the paper is well written. Some parts may need some extra text or explanations:

- Page 6, line 100-101. Please add how RPM is defined.

- Page 6, line 100-101. It is unclear why a cut off of 1000 RPM or 1 RPM is taken for unknown and known viruses respectively. Some clarification is needed.

- Page 6 line 114-117. Most probably the E.coli reads in figure 2, in cases and controls, are contaminations or reads originating from library-prep ingredients. Presumably the reads were also found in blank controls, and the manuscript would improve if presence of E.coli in all samples (and blancs?) is mentioned, and explanation is added.

- Page 14, lin 284: samples from healthy volunteers: Please include in which month of which year the control healthy persons were sampled. Was this also in 2016-2017? If sampled in a different year, then this could be mentioned in the discussion as a weakness in the study design

- Page 15, line 303-304. RPM: it is not clear what the reads per million exactly represents (per million of what?). Since cut-offs are based on RPMs this needs clear explanation

- Page 15 line 321. Table SX, is probably Table S1

Reviewer #2: The manuscript by Shi et al is a traditional viral metagenomics study looking at BALF samples from 408 respiratory cases and 37 healthy controls collected 2016-17 from Wuhan. It is expertly done including analyzing total RNA including those of viruses, bacteria, and fungi. Mostly traditional human viral pathogens are identified as well as bacterial and fungal pathogens. Of note a zoonotic infection with an avian bacterial pathogen was detected as well as enterovirus D68. The wide range of viruses found is interpreted as reflecting Wuhan's connectivity with rest of China and the world.

Reviewer #3: The scientific goal of this research (and thus the value of the findings) is not clear. There is no reason to suspect that SARS-CoV-2 could be circulating in humans before 2019. Thus, the value of a finding that it was absent in Wuhan in 2016 is not clear. The findings regarding infectome in pneumonia patients are not referred to infectomes elsewhere in the world or to any reference groups. There is no hypothesis how the infectome in 2016 could be linked to the SARS-CoV-2 emergence or the outbreak spread. A description of infectome in pneumonia patients may be interesting by itself, but has limited scientific impact and is more suitable for journals that do not consider scientific impact. The phylogenetic analysis is ridiculous. While the methodology of tree reconstruction is acceptable, it was not detailed how a few dozen sequences out of thousands available in Genbank were chosen for the analysis. Without a clear and well explained reference sequence selection algorithm, it is not possible to judge the validity of conclusions. The reviewer does not see how this manuscript could be improved to make it acceptable for publication in PLoS Pathogens. However, if unsupported claims (first of all, the phylogenetic analysis) are omitted or explicitly resolved, and reference to SARS-CoV-2 is removed (the data are not at all related to the new CoV), the analysis may be publishable in a more general journal, such as PLoS One.

Reviewer #4: In this study, Shi et al. examined pneumonia and acute respiratory infections at the Central Hospital of Wuhan between 2016 and 2017 by sequencing the bronchoalveolar lavage fluid samples from 408 patients. They identified 37 pathogen species, comprising RNA viruses, DNA viruses, bacteria and fungi, defining a stable core infectome. The prevalence of an atypical respiratory virus EV-D68, and a single case of a sporadic zoonotic pathogen – Chlamydia psittaci was reported. These data also suggest that SARS-CoV-2 or closely related viruses were absent from Wuhan in 2016-2017. It is an important study in the context of investigating early events of COVID-19 transmission in Wuhan. But the analysis and presentation need further improvement.

**Part II – Major Issues: Key Experiments Required for Acceptance**

Reviewer #1: None

Reviewer #2: Technically this study is well performed and issues such as contamination and index hopping are appropriately addressed and the conclusions well supported.

31 No SARS CoV2 RNA was detected but because NGS is generally less sensitive than a good RT-PCR the authors should discuss whether al the RNAs were also analyzed by qPCR. It is conceivable that a still poorly replicating SARS2 precursor was replicating in human but a low viral loads not detectable by NGS. It seems it would have been easy to add that qPCR to the long list of qPCR already used used.

Reviewer #3: None

Reviewer #4: Major points:

1. Line 164 and Figure 4: What is the rationale of choosing the viral sequences to be presented on the phylogeny tree? Were they well-characterized or circulating in the world previously or during the time of sampling (2016 to 2017)? Since the viral sequences themselves can determine the tree, if there are not enough sequences from different geometric regions, the trend of HCoVs clustering with US sequences is not conclusive. It can simply be due to more US sequences in the database. A similar trend can be found in IVA and HPIV1 in Figure 4.

2. The infectome analysis was not thorough enough. For example, 1) no genome coverage and depth was shown for any pathogen NGS results which can be more storytelling than RPM value alone. 2) Any correlation of age, sex, severity and pre-existing condition to the time of sampling or between each other? Any correlation of pathogens identified with age/sex/severity/pre-existing conditions?

3. The key focus was understantably on SARS-CoV-2. But did the authors also examined the presence of other SARS-related CoVs? It will be useful to state/discuss this.

**Part III – Minor Issues: Editorial and Data Presentation Modifications**

Reviewer #1: See bulletpoints in the summary

Reviewer #2: 27 Would be interesting to hear whether or not such BALF were collected later than 2017.

Introduction: 72 Surely there are several prior human metagenomics studies of respiratory diseases that have been published and could be referred and compared to.

https://pubmed.ncbi.nlm.nih.gov/32497137/

https://pubmed.ncbi.nlm.nih.gov/33367583/

https://pubmed.ncbi.nlm.nih.gov/32872469/

etc

147 BLAF typo

153 Strikingly, strong correlations…. Plenty of prior publications have reported on this correlation. Include a few references here or in discussion.

232 change 106 to 106

Clarify whether he confirmatory PCR were done on only the samples found positive by NGS or on all cDNAs? If so were any infections missed by NGS but seen by qPCR?

Reviewer #3: None

Reviewer #4: Minor points:

1. Line 94: For each patient, was the BALF sample taken upon hospitalization or some time afterwards? Clarify this as it would be important for the timing of NGS.

2. Line 107: Please clarify co-infection. Should it be two different pathogens (for example, two different RNA viruses) or two different category pathogens (for example, one virus and one bacteria)?

3. Line 124 and Figure 2C: For each category(RNA/DNA Viruses, Bacteria and Fungi), what is the order for each pathogen? Some are ranked alphabetically (Bacteria), which makes no sense. Should be ordered according to the taxonomy (Order or Family) or prevalence.

4. Line 151: “For each of the pathogens identified here”, please clarify “here”. Does it mean all 37 pathogens were validated through qPCR, and were all these results shown?

5. Figure 1C: The Y-axis should labeled “Age” and not “No of cases”.

6. Figure 2A: Actual number and fraction not shown. Please show this information as a pie chart alone can be visually misleading.

7. Figure 2B: Please show the actual number of cases for each bar.

8. Figure 3: Please revise the x-axis range for IVA and HPIV3. A Ct value of less than 10 and above 40 is impractical. There are no qPCR results shown for bacteria and fungi, but Table S1 indicated so. Should show at least one for each category.

9. Figure 5: The figure legend indicated only non-zero abundance levels are reported, but zero was actually shown on the figure for control groups (also according to the heatmap in Figure 2C). Is this non-zero abundance is only applied to the patient group?

10. Line 281: 409 or 408 samples?

11. Line 296-298: Version of the software and database not shown. What is the software used for the mapping?

PLOS authors have the option to publish the peer review history of their article (what does this mean?). If published, this will include your full peer review and any attached files.

Reviewer #1: No

Reviewer #2: No

Reviewer #3: No

Reviewer #4: No
---

## [Decision Letter · Decision Letter 1]

8 Jan 2022

Dear Dr. Zhang,

We are pleased to inform you that your manuscript 'Total Infectome Characterization of Respiratory Infections in pre-COVID-19 Wuhan, China' has been provisionally accepted for publication in PLOS Pathogens.

Before your manuscript can be formally accepted you will need to address a minor issue specified below and to complete some formatting changes, which you will receive in a follow up email. A member of our team will be in touch with a set of requests.

Best regards,

Alexander E. Gorbalenya, PhD, DSci

Section Editor

PLOS Pathogens

Marco Vignuzzi

Section Editor

PLOS Pathogens

Kasturi Haldar

Editor-in-Chief

PLOS Pathogens

orcid.org/0000-0001-5065-158X

Michael Malim

Editor-in-Chief

PLOS Pathogens

orcid.org/0000-0002-7699-2064

Please address the minor issue raised by the Reviewer #2 in the final version.

Reviewer Comments (if any, and for reference):

Reviewer's Responses to Questions

**Part I - Summary**

Reviewer #2: resubmission addressed all my concerns

Reviewer #4: NA

**Part II – Major Issues: Key Experiments Required for Acceptance**

Reviewer #2: (No Response)

Reviewer #4: NA

**Part III – Minor Issues: Editorial and Data Presentation Modifications**

Reviewer #2: Minor point needs clarifying: The EV-D68 cases identified here showed moderate to severe respiratory symptoms, although viral abundance was generally very high, with four of six cases showing >106 RPM (i.e., >10% of total RNA) in the BALF sample.

It is not clear how 106 (Is that 106 OR 1 million?) constitute 10% of total RNA. 106 RMP would be 106/1000000 or 0.000106 (0.016%) of all reads not 10%.

Reviewer #4: All issues raised by this reviewer were addressed satisfactory in the revised manuscript.

PLOS authors have the option to publish the peer review history of their article (what does this mean?). If published, this will include your full peer review and any attached files.

Reviewer #2: No

Reviewer #4: No

---

## [Editor Report · Acceptance letter]

25 Jan 2022

Dear Dr. Zhang,

We are delighted to inform you that your manuscript, "Total Infectome Characterization of Respiratory Infections in pre-COVID-19 Wuhan, China," has been formally accepted for publication in PLOS Pathogens.

Best regards,

Kasturi Haldar

Editor-in-Chief

PLOS Pathogens

orcid.org/0000-0001-5065-158X

Michael Malim

Editor-in-Chief

PLOS Pathogens

orcid.org/0000-0002-7699-2064